# Determinants of Safe Pesticide Handling and Application Among Rural Farmers

**DOI:** 10.3390/ijerph22020211

**Published:** 2025-02-02

**Authors:** Olamide Stephanie Oshingbade, Haruna Musa Moda, Shade John Akinsete, Mumuni Adejumo, Norr Hassan

**Affiliations:** 1Department of Health Professions, Faculty of Health and Education, Manchester Metropolitan University, Manchester M15 6BG, UK; oshingbadeolamide@gmail.com; 2Department of Environmental Health and Safety, University of Doha for Science and Technology, Doha P.O. Box 24449, Qatar; norr.hassan@udst.edu.qa; 3Department of Environmental Health Sciences, University of Ibadan, Ibadan P.O. Box 4078, Nigeria; sjohnadisa@yahoo.com (S.J.A.); adejumo_mumuni@yahoo.com (M.A.)

**Keywords:** farm pest control, exposure, hazard awareness, developing countries

## Abstract

The study investigated the determinants of safe pesticide handling and application among farmers in rural communities of Oyo State, ssouthwestern Nigeria. A cross-sectional design utilizing 2-stage cluster sampling techniques was used to select Ido and Ibarapa central Local Government Areas and to interview 383 farmers via a structured questionnaire. Data were analyzed using descriptive statistics and logistic regression at *p* = 0.05. Results showed that 41.8% of the farmers had been working with pesticides on farms for at least 5 years, 33.0% attended training on pesticide application, 73.5% had good safety and health knowledge, and 72.3% had safe pesticide handling and application practices. About half (50.2%) stated that they wear coveralls, gloves, and masks to protect their body, face, and hands when applying pesticides, 9.8% use empty pesticide containers for other purposes in the house/farm, while 11.5% blow the nozzle with their mouth to unclog it if it becomes blocked. The three major health symptoms reported by the participants were skin irritation (65.0%), itchy eyes (51.3%), and excessive sweating (32.5%). Having attended training on pesticide application and use enhanced (OR = 2.821; C.I = 1.513–5.261) practicing safe pesticide handling and application. Farmers with good knowledge (OR = 5.494; C.I = 3.385–8.919) were more likely to practice safe pesticide handling and application than those with poor knowledge about pesticide use. It is essential to develop and deliver mandatory comprehensive training programs for farmers on impacts of pesticides on health and environment, along with sustainable safe handling, application, and disposal of pesticides using proper waste management techniques and recognizing early signs and seeking medical assistance. The urgent need to strengthen policy to regulate pesticide use and limit farmers’ access to banned products is also key.

## 1. Introduction

Pesticide application plays a crucial role in modern agriculture, enhancing crop yields, and ensuring food security, especially in rural communities where agriculture is the primary livelihood. Correspondingly, poor handling of pesticides presents significant risks to the health of farmers, the crop itself, and the environment, thus making it essential to enhance farmers’ pesticide safety and health handling practices [1]. Pesticides have become an indispensable chemical used among farmers to prevent pre- and post-harvest losses, ensuring sustainable food production by increasing yields and providing year-round food availability [2,3,4,5] partly due to climate change impacts and increased pest resistance to other control methods. Around 3.5 million tons of pesticide have been used globally in 2021 with a marginal increase in its use in Africa between 2020 and 2021 from 203 kt to 204 kt [6]. Nigeria is among the largest importer of pesticides in the continent by volume with a reported 147,477 tons imported in 2020 [7]. Despite its increased use in farming activity, only around 1% of these products are effectively employed in the control of farm pests, leaving the remaining as residue in other secondary media, thereby resulting in harm to both humans and the environment [8]. Weak regulation across the continent, in addition to lack of awareness of farmers, led to the importation and use of globally banned pesticides [9]. A recent study revealed that 80% of pesticides used among small-scale farmers in Nigeria are classed as a highly hazardous group according to WHO, thereby contributing to the rise in non-communicable disease rates and deaths within farming communities [10]. The unregulated nature of pesticide access among farmers, its availability in the market, and farmers’ trust in informal information sources in most developing countries are factors associated with the persistent and extensive use of these toxic pesticide classes [11,12], thus resulting in deleterious effects on the environment and increased human health concerns [13].

Skin contact is a commonly reported pesticide exposure route; however, inhalation due to poor safety practices has also been reported and can lead to both acute toxicity effect and chronic disease development [8,14,15,16]. The most common pesticide exposure-related health effects reported in earlier studies include headache, skin irritation, eye irritation, fatigue, and muscle pain [12,15,17]. The misuse of pesticides can result in secondary pest outbreaks [18], extinction of non-target species [13], soil, water, and air contamination [19], accumulation of residues in primary and derived agricultural products, and other associated hazards to humans and the environment [20]. Primarily, farmers are at a greater risk of pesticide exposure due to pesticide residues in treated crops, handling, storage and disposal practices, maintenance of application equipment, and lack or non-use of personal protective equipment properly [11,12,14,21]. Lack of knowledge regarding pesticide hazards [22], low farmer awareness and attitudes toward pesticide exposure risk [23], and lack of education and understanding of safe pesticide handling practices that including storage and disposal [1,24,25,26], and not using the appropriate equipment while spraying pesticides [8], are precursor to high pesticide exposure among farmers and pesticide residues on crops [19].

The major obstacles to farmers’ adoption of self-protection behavior, particularly the use of personal protective equipment, have been identified as farm safety illiteracy and ignorance regarding the amount to which pesticides pose a threat [27,28]. There is inadequate information regarding these factors influencing safe pesticide handling and application among farmers in rural communities, particularly in Nigeria. Understanding farmers’ knowledge of pesticides and the adoption of safety precautions are critical to identifying exposure situations and knowledge gaps and providing information to help inform farmers and the development of policy to prevent or reduce health and environmental risks associated with pesticide use [28]. In this study, the authors assessed determinants of safe pesticide handling and application practices among farmers in rural communities of Oyo State, southwestern Nigeria.

## 2. Materials and Methods

### 2.1. Study Location

The research took place in two sites, both located in Oyo State, Nigeria. Ido Local Government Area (LGA) has a population of 104,261 residing within a 1016.95 km^2^ landmass and has tropical climatic conditions that support rainforest vegetation. The community location within the deciduous forest in the central part of Oyo State makes it one of the most viable areas for agriculture. Its inhabitants are predominantly farmers cultivating both food and cash crops on a small and commercial scale. Ibarapa Central Local Government Area is the second location and has a population of 103,243 inhabitants with a landmass of 440 km^2^ [29]. It shares a boundary with Ogun State to the south and west, Ibarapa North Local Government Area to the north-west, Iseyin Local Government Area to the north-east, and Ibarapa East Local Government to the east [30]. Most of the residents engage in agriculture due to abundant fertile farmlands. Yam tubers, cassava, mangoes, cashew, palm kernel, corn millet, melon, tomatoes, okro, and cocoa are some of the major crops produced in these lands in large quantities for local consumption and even for export [31].

### 2.2. Study Design and Study Population

Using a cross-sectional design, 383 farmers constituting 85% of the invited participants involved in farming activities for at least one year and in the purchase, use and storage of pesticides in the two selected LGA, were interviewed using a structured questionnaire. Farmers below the age of 18 years and those with limited years of experience in pesticide handling and application were excluded from the study. The exclusion of these groups was made in order to comply with the age of consent and also ensure the sampled group had sufficient knowledge and experience of pesticide application. Due to a lack of official population records regarding individuals whose primary source of income is classed as farming, an estimate of the active population involved in farming activity was set at 100,000 individuals within the two LGAs, and a minimum sample size of 383 was estimated as sufficient to support the study [11,12,21].

### 2.3. Sampling Procedure

Two-stage cluster sampling techniques were used to select two rural Local Government Areas (Ido and Ibarapa Central) in Oyo State and 383 volunteered farmers participated in the study. Villages within Ido Local Government Area (600 villages) and Ibarapa Central LGA (300 villages) were grouped into 60 clusters for each LGA. From these clustered villages, ten clusters were randomly selected from each LGA, resulting in 20 clusters in total (10 from Ido and 10 from Ibarapa Central). A total of 383 participants who consented to the study were selected from these clusters, with a refusal rate of 4.25%.

To encourage farmers’ participation in the study, the researchers approached respective farmers’ association executives in the selected LGAs to help raise the study awareness among the target group. A further meeting was held with the farmers within the study area to discuss the study theme and objectives of the study and address any concerns regarding the study. Written consent was obtained from each participant prior to their inclusion in the questionnaire’s interview process.

### 2.4. Data Collection Tools Procedure

A closed-ended structured questionnaire was used to collect data on farmers’ socio-demographic characteristics, occupational history, pesticide safety knowledge (12-point scale), safety attitude about pesticide handling and application (9-point scale), and safety practices during pesticide handling and application (11-point scale). Pesticide safety knowledge refers to the understanding farmers have about the safe use of pesticides, including safety measures, proper storage, correct application methods, and handling of pesticides to minimize risks to health and the environment. The knowledge was measured using 12-item structured questions. Some of the items were “Pesticide does affect human health”, “Pesticide does affect livestock health”, “Pesticides are essential for high crop yield”, etc. Thereafter, each of the items was assigned 1 point making a total of 12 points. Knowledge score was rated as poor (scores < 6) and good (scores ≥ 6).

Safety attitude about pesticide handling and application refers to the perceptions and beliefs rural farmers have regarding the importance and effectiveness of safe pesticide handling. The attitude was assessed using 9 items on a Likert scale (from “Strongly Agree” to “Strongly Disagree”). Some of the items include “Smoking during pesticides application increase chance of its entrance to the body”; “Drinking/eating while handling pesticide increase potential entrance to the body”; “Personal protective equipment (PPE) is important to prevent the body from pesticide poisoning”, etc. Thereafter, points were assigned to each of the items; attitude score was computed and categorized as negative attitude (scores < 5) and positive attitude (scores ≥ 5), respectively.

Safety practices during pesticide handling and application refer to the actual actions that rural farmers take in handling and applying pesticides safely. Practices were measured using 11 items on a structured questionnaire. The items include “I regularly use sprayer during pesticide spraying/application on my farm”; “Use empty pesticide container for other purposes/use in the house/farm”; “I purchase pesticides sufficient for one cropping season”; “I wear gloves and mask to protect my face and hands when applying pesticides on farm”; “I wear coverall/farm uniform when applying pesticides on farm”; “I always read the safety instruction on the pesticide container before use”; “I blow the nozzle with mouth to unclog out if it gets blocked”, etc. Each of the items was assigned 1 point, making a total of 11 points, and scores were rated as unsafe (scores < 6) and safe (scores ≥ 6) practices.

Interviews were conducted by six trained Research Assistants who had completed tertiary education and were acquainted with interview data collection techniques. Interviewers were trained on how to use the questionnaire and how they should introduce themselves and the research objectives, with modesty, to the farmers during the interview. The questionnaire was developed in English and later translated into the local Yoruba language by a professional translator before being administered. Each interview lasted around 30 min.

### 2.5. Data Analysis

Data collected were checked for completeness, coded, and analyzed using the statistical package for social sciences (SPSS, version 22.0). The analyzed results were presented in mean standard deviation for continuous variables and percentages for categorical variables. Chi-square test was used to analyze the association between respondents’ pesticide handling and application practices category and socio-demographic characteristics, knowledge, and attitude about pesticide use. Ordinary logistic regression analysis was carried out to measure the influence of respondents’ educational status, smoking habits, access to training, knowledge, and attitude on safe pesticide handling and application practices. Statistical significance was defined at *p* = 0.05.

### 2.6. Ethical Consideration

Ethical approval to conduct this research was obtained from Manchester Metropolitan University, Department of Health Professions Ethics Committee number: 53061 before the commencement of the fieldwork. Anonymity and confidentially were assured to participants, and individual consents were obtained before taking part in the interview.

## 3. Results

Table 1 presents the socio-demographic characteristics of the participants. From the result, 34.0% of the participants were within the age of 41–45 years, 27.3% were 31–35 years old, 84.3% were male, 43.8% had completed secondary education, while 15.8% had attained tertiary education. The majority (77.5%) of farmers reported that they have never smoked, while 10.5% smoke regularly. Several (41.8%) of the participants had been working with pesticides on farms for about 5 years, 58.5% stated that they usually work for half a day on the farm, and 33.0% said they had attended training on pesticide application. The majority (86.0%) of the participants stated that they practice farming during the wet and dry seasons, 43.3% described that they usually apply pesticides twice per crop growing season, 32% apply it three times, while 21.5% apply even more. Out of those interviewed, 48.5% were landowners and 68.0% had more than one hectare of farmland, as presented in Table 1.

### 3.1. Safety and Health Knowledge and Attitude About Pesticide Handling and Application

Results of the respondents’ safety and health knowledge regarding pesticide handling are presented in Table 2. Based on the study outcome, 95.3% of the respondents affirmed pesticide exposure does affect human health. Similarly, 92.8% acknowledge pesticides affects livestock health, while 86.3% opined frequent use of pesticide does present an environmental impact in the long run. As part of the rationale behind the frequent use of pesticides as a method of farm pest control, 93.5% of the respondents considered it essential for high crop yield. A significant percentage of respondents (82.8%) stated that they possess the right skills and knowledge to safely apply pesticides on the farm partly due to frequent use of the chemical over years.

Respondents’ safety and health attitude toward pesticide handling, application, and storage revealed that 10.5% engage in smoking during pesticide application thereby increasing the chance of ingesting active ingredients contained in the formulation. Another poor hygiene and safety behavior affirmed by the respondents showed 87.1% have at some point drunk/eaten while handling pesticides, thus resulting in increasing the chances of entrance to the body. The use of personal protective equipment was considered an important means of reducing the chance of pesticide exposure during handling and application by 88.3% of the farmers. The comfort of PPE use during the application period was considered okay by 59.5% who agreed that by wearing the required PPE will not slow them down; however, 15.6% of respondents found the equipment as inconvenient and preferred not to use it during pesticide application. Environmental awareness related to pesticides found that 75.8% disagreed with washing sprayer tanks in a river or waterway due to its associated impact on the ecosystem. Relatedly, health concern was regarded as high, with 85.8% affirming their disagreement with the mixture of decanted pesticides by hand in sprayer tanks (Table 2).

### 3.2. Safety and Health Practices About Pesticide Application

Table 3 presents respondents’ related practices during pesticide handling and application. From the results, 95.3% of farmers sampled indicated that they currently use pesticides on their crops and 96.8% said they use mechanical sprayers due to their convenience and efficiency. A review of the empty container disposal method adopted among the respondents revealed some environmental and health concerns based on affirmed practices among 9.8% of the respondents who affirmed at some point ever used empty pesticide containers for other purposes either on the farm or in their houses while negating the safety instructions as prescribed on the safety data sheet and container label. Storage of pesticides at home is a common practice among the respondents with 79.5% affirming this practice although they claim the chemicals are stored in a safe and secured location as against stored on the farm. The use of gloves and masks to protect and wearing a coverall/farm uniform when applying pesticides on a farm was found to be a common practice only among half (50.5%) of the respondents. Access to safety information inquired among the participants showed that 58.5% read either safety instructions supplied on the pesticide container or the accompanied safety data sheet prior to its use. Those who follow label directions carefully for preparation and application of pesticides were 65% and those who disposed of empty containers according to the after-use instructions were only 54.3%. The tendency for the nozzle of the sprayer to become clogged during use was another source of safety concern considered in the study where 11.5% said they blow the sprayer nozzle with their mouth to unblock the nozzle. From the reported health symptoms associated with pesticide intoxication, three major symptoms reported by the participants were skin irritation (65.0%), itchy eyes (51.3%), and excessive sweating (32.5%) which further raises concern about the safe use of PPE as claimed earlier among the respondents (Figure 1).

### 3.3. Category According to Socio-Demographic Characteristics and Knowledge

Distribution of respondents’ handling and application practices score category according to their socio-demographic characteristics, knowledge, and attitude about pesticide use was measured (Table 4). The results showed no significant association between respondents’ age categories, the duration of pesticide handling, and their scores on pesticide handling and application practices. Significantly, respondents who had completed tertiary education (92.1%) practiced safe pesticide handling and application compared to their counterparts with lower educational attainment (*p*-value < 0.001). Respondents (76.5%) who never smoke significantly practiced safe pesticide handling and application compared to those who were either smokers or had quit smoking (*p*-value = 0.001). Respondents (87.9%) who had received training on pesticide application significantly practiced safe handling and application compared to those who have not received training (*p*-value < 0.001). Significantly, respondents (84.4%) who applied pesticides twice on their farm, and those with good knowledge (82.0%) and positive attitudes (86.6%) toward pesticide application, both significantly practiced safe handling and application, respectively (*p*-value < 0.001). A survey of the most commonly used pesticides among the respondents revealed that 54.8% of respondents preferred Cypermethrin (Best), a WHO class II pesticide, as the most preferred chemical of choice. Another WHO Class II pesticide, Lamda-cyhalothrin (Lara Force), is used among 37.9% of the respondents. Dichlorvos (Sharpshooter), WHO Class I pesticide, was confirmed to be used by 2.8% of respondents on their farms.

Table 5 presents an ordinary logistic regression analysis of the respondents’ educational status, smoking habit, access to safety training, knowledge, and attitude influencing their safe handling and application practices. Data analyzed showed farmers with tertiary education (OR = 8.082; C.I = 2.625–24.870) were more likely to practice safe pesticide handling and application than those with lower or no formal education. In addition, participants who have had training on pesticide application and use presented better safe pesticide handling practices (OR = 2.821; C.I = 1.513–5.261). Relatedly, farmers with good knowledge (OR = 5.494; C.I = 3.385–8.919) were more likely to practice safe pesticide handling and application than those with poor knowledge about pesticide use. Additionally, a positive attitude toward pesticide use does influence (OR = 6.624; C.I = 4.083–10.748) safe handling and application practices among the respondents (Table 5).

## 4. Discussion

Pesticide use and handling practices in several communities have increased as a means of protecting farm produce to boost food security and help in meeting related sustainable development goals and community food security. Relatedly, farmers are increasingly exposed to increased poisoning and the development of pesticide illnesses that can be chronic in nature. Due to the nature of the work and the prevailing environmental condition, inhalation of pesticides during spraying, dermal contact associated with lack of PPE use, and accidental ingestion through contaminated food or water are major potential routes of exposure. The need to raise awareness of these risks among farmers and farm workers and highlight the importance of responsible pesticide use and adoption of safety measures and practices is essential to minimize related health hazards [1,11,32,33]. On this premise, assessed farmers’ knowledge, attitude, and practices regard the handling and application of pesticides among rural communities of Oyo State, southwestern Nigeria, revealed the need for closer attention by stakeholders to help raise awareness regarding safety precautions needed when handling the product.

The outcome from the study revealed the preferred pesticides of use among the sampled group are either classed as highly hazardous (Class I) or moderately hazardous (Class II) according to WHO classification [34]. These further raises health and safety concerns around the potential exposure of farmers to highly hazardous compounds both on farms and in homes where there is a greater risk of the chemicals being stored or transported along with other materials. The relatively low level of education among the sampled farmers may be one of the reasons behind their inability to read pesticide product labels written in English, and thus not understanding the safety measures to apply when handling these highly hazardous pesticide (HHP) compounds.

To encourage safe farm practices, farmers’ formal education, training, and awareness are vital as they help with shaping safety habits, enhance critical thinking, and problem-solving skills alongside an increased tendency to apply required controls to help mitigate risk and/or improve safety consideration [15,24,35]. The outcome of the study showed that farmers who have received a higher form of formal education and/or had previous training on pesticide application demonstrate a higher tendency to adhere to safety practices considering that they are most likely to be aware of potential hazards associated with pesticide use and the importance of adhering to safety guidelines that include the proper use of personal protective equipment (PPE), safe handling, storage, and disposal of pesticides [1,36]. In line with this, multiple studies have associated training with the increased comprehension among farmers of pesticide hazard management [32,37,38]. To help manage frequent pesticide use and integrated pest management, a more healthy and environmentally friendly approach could be considered alongside pesticide safety literacy [39]. Years of handling pesticides on the farm offers another path whereby respondents gain further information regarding the safe handling of the product. This is consistent with the reports of previous studies that longer years of pesticide handling experience are more likely to demonstrate safe methods of pesticide application and handling [26,40,41].

Smoking was found to be significantly associated with safety habits and practices among the sampled group. Farmers who had never smoked had better safety habits and practices compared to those who were current or previous smokers. In addition, there was high respondents’ agreement regarding the lack of PPE use, and drinking and eating while handling pesticides with increased potential of pesticide ingestion. This further corroborates on previous studies that identified poor personal hygiene and safety practices such as limited use of coveralls and respirators (PPE), smoking during application, and eating kola nuts among the farmers, as pathways through which farmers are exposed to pesticide residue [12,42].

There is a greater agreement among the respondents concerning unsafe disposal of empty pesticide containers and washing of pesticide spraying containers in nearby surface water such as lakes and streams, impacting water quality and harming the environment, which aligns with a previous study result [12]. Landfilling can be considered as a viable way of disposal; however, clearly farmers have insufficient information about this, hence a proper training program on waste management should be developed and delivered. A proportion of the respondents revealed that they blow the nozzle with their mouth to unclog it if it becomes blocked. This is consistent with a study that reported blowing a clogged nozzle with the mouth, talking while spraying pesticides, and mixing pesticides with bare hands [12]. This can be explained by the lack of knowledge on pesticide toxicity and the hesitation in following label instructions. More than half stated that they always read the safety instructions on the pesticide container before use. A high percentage reported that they follow label directions carefully for preparation and application of pesticides.

Studies have reported that poor pesticide handling practices may expose the farmers to a number of pesticide-related health symptoms [43,44]. Short- and medium-term post-pesticide exposure symptoms reported by farmers included skin irritation, itchy eyes, excessive sweating, poor vision, fatigue/weakness, and others. These symptoms have also been reported among banana plantation farmers in Ecuador [45], cacao farmers in southwestern Nigeria [46], and rural farmers in northern Nigeria [11,12]. These underscore the significance of direct skin contact with pesticides during handling or spraying activities. Pesticide formulations can contain chemicals that may irritate the skin, leading to various skin conditions, and the eyes are also sensitive to pesticide fumes or residues. Impaired vision may also arise from accidental pesticide exposure to the eyes or face [11,12]. This demonstrates the urgency to enhance farmers’ awareness to recognize early symptoms of pesticide exposure [47]. In addition, another important issue that should be raised among farmers is the need to quickly seek medical assistance once they recognize their symptoms to avoid further health complications. A study showed that in three different countries farmers often believe that pesticide-related symptoms are normal and therefore do not seek medical treatment [48].

These findings highlight the multifaceted nature of pesticide-related health risks, underscore the importance of comprehensive safety measures, and emphasize the need for farmers to use appropriate personal protective equipment (PPE) to minimize skin exposure and protect themselves from potential skin-related health problems. The study found that there was no significant association between respondents’ age category and category of respondents’ handling and application practices score. However, respondents who had completed tertiary education significantly practiced safe pesticide handling and application compared to their counterparts with lower educational attainment. Previous studies also alluded to this result where the level of education of the farmer has a significant association with their willingness to practice safe pesticide application on their farms [15,25,26,38,49]. This may be associated with farmers’ ability to read and willingness to adopt new technology and skills to enhance safe application practices.

## 5. Conclusions

The authors acknowledge the study was limited to two Local Government Areas and thereby calls for caution around the generalization of farmers’ safety behaviors in Oyo State and Nigeria at large. In addition, the adoption of the cross-sectional method is susceptible to bias that includes non-response and recall bias.

However, it is evident from the participants’ response, that the need to boost crop yield and enhance food security are important factors for the frequent pesticide use among farmers which presents potential health and environmental impacts. Farmers’ safe use of pesticides is critical in managing human acute or chronic diseases and environmental pollution. The finding from the current study raised concern about an increased risk of highly hazardous pesticide (HHP) exposure and pesticide toxicity due to its acceptance as a pesticide of choice. The long-term health consequence in relation to frequent use of HHPs alongside poor use of personal protective equipment among the respondents is a call on the regulatory bodies, including governmental and non-governmental organizations, to enhance pesticide safety education, conduct mandatory training programs, and design strategies and action plans among farmers, in order to mitigate against diseases among farmers. The training should include information on different types of pesticides, an integrated pesticide management system, safe handling, safe disposal, and safe storing of pesticides. Also, the training should cover the importance of and how to use PPE, how to observe personal hygiene, recognize early symptoms, and where and how to seek medical assistance. In addition, considering the use of HPPs, translating labels to the native language, and the need to strengthen policy aimed at regulating pesticide use will go a long way to limit farmers’ access to banned products.

## Figures and Tables

**Figure 1 ijerph-22-00211-f001:**
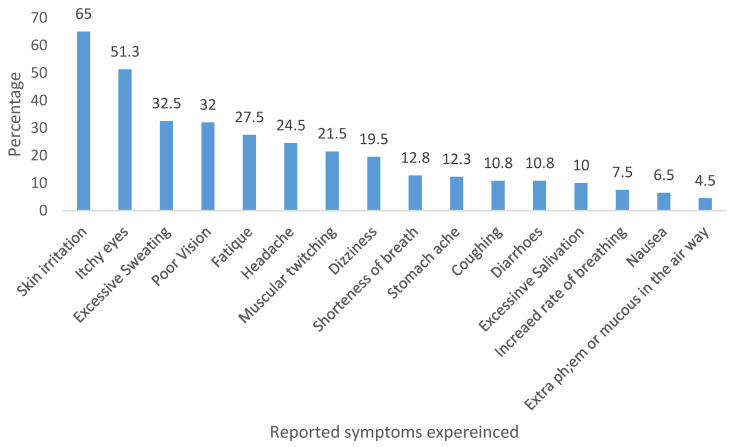
Associated health impacts of incorrect pesticide handling and application.

**Table 1 ijerph-22-00211-t001:** Socio-demographic characteristics of the study group.

Characteristics	Frequency (%)
Age	
18–25	27 (6.8)
26–30	57 (14.3)
31–35	109 (27.3)
36–40	14 (3.5)
41–45	136 (34.0)
46 and above	57 (14.3)
Gender	
Male	337 (84.3)
Female	63 (15.8)
Highest level of education attained	
No formal	53 (13.3)
Primary	109 (27.3)
Secondary	175 (43.8)
Tertiary	63 (15.8)
Smoking habit	
Smokers	42 (10.5)
Never smoked	310 (77.5)
Quit smoking	48 (12.0)
Number of years working with pesticides on farm	
1–5 years	167 (41.8)
6–10 years	155 (38.8)
11–15 years	60 (15.0)
16–20 years	18 (4.5)
Attended training on pesticide application	132 (33.0)
Work shift	
Full day	166 (41.5)
Half day	234 (58.5)
Season farming practices	
Wet and dry	344 (86.0)
Only wet season	43 (10.8)
Only dry season	13 (3.3)
Time of pesticide application per crop growing season	
Once	13 (3.3)
Twice	173 (43.3)
Thrice	128 (32.0)
More than three times	86 (21.5)
Farm size	
≤1 hectare	128 (32.0)
>1 hectare	272 (68.0)

**Table 2 ijerph-22-00211-t002:** Safety and health knowledge and attitude about pesticide handling and application.

Statement	Frequency (%)
Knowledge statement	
Pesticide does affect human health.	381 (95.3)
Pesticide does affect livestock health.	371 (92.8)
Pesticide does affect environment.	345 (86.3)
Pesticides are essential for high crop yield.	374 (93.5)
Know how to safely apply or spray pesticides on the farm.	331 (82.8)
Attitude statement	
Smoking during pesticide application increases chance of its entrance to the body.	
Agree	288 (72.0)
Disagree	28 (5.6)
Drinking/eating while handling pesticide increases potential entrance to the body.	
Agree	348 (87.1)
Disagree	28 (7.1)
Personal protective equipment (PPE) is important to prevent the body from pesticide poisoning.	
Agree	353 (88.3)
Disagree	16 (4.0)
Using PPE could slow someone down during pesticide application.	
Agree	62 (15.6)
Disagree	238 (59.5)
Sprayer tanks can be washed in a river or waterway without any damage to the ecosystem.	
Agree	50 (12.5)
Disagree	303 (75.8)
Pesticides can be mixed with naked hand.	
Agree	44 (11.2)
Disagree	343 (85.8)

**Table 3 ijerph-22-00211-t003:** Pesticide handling and application practices.

Practices	Frequency (%)
Currently use pesticides on my crops.	381 (95.3)
Regularly use sprayer during pesticide spraying/application on my farm.	387 (96.8)
Use empty pesticide container for other purposes/use in the house/farm.	39 (9.8)
Purchase pesticides sufficient for one cropping season.	173 (43.3)
Store pesticides at home in safe and secured locations.	318 (79.5)
Wear gloves and mask to protect my face and hands when applying pesticides on farm.	202 (50.5)
Wear a coverall/farm uniform when applying pesticides on farm.	211 (52.8)
Always read the safety instructions on the pesticide container before use.	234 (58.5)
Blow the nozzle with mouth to unclog it if it becomes blocked.	46 (11.5)
Wash contaminated farm cloth separately after using pesticides on the farm.	354 (88.5)
Dispose of empty containers according to the after-use instructions.	217 (54.3)
Follow label directions carefully for preparation and application of pesticides.	260 (65.0)

**Table 4 ijerph-22-00211-t004:** Percentage distribution of respondents’ handling and application practices score category according to their socio-demographic characteristics and knowledge about pesticide use.

Socio-Demographic Characteristics	Practices Score Category			χ^2^ Fisher’s Exact (*p*-Value)
	Unsafe (%)	Safe (%)	Total (%)	
Age in years				
18–25	7 (25.9)	20 (74.1)	27 (100)	2.137 (0.830)
26–30	12 (21.1)	45 (78.9)	57 (100)	
31–35	31 (28.4)	78 (71.6)	109 (100)	
36–40	3 (21.4)	11 (78.6)	14 (100)	
41–45	41 (30.1)	95 (69.9)	136 (100)	
46–50	17 (29.8)	40 (70.2)	57 (100)	
Education				
No formal education	26 (49.1)	27 (50.9)	53 (100)	27.305 (<0.001)
Completed primary school	37 (33.9)	72 (66.1)	109 (100)	
Completed secondary school	43 (24.6)	132 (75.4)	175 (100)	
Tertiary	5 (7.9)	58 (92.1)	63 (100)	
Smoking Habit				
Smokers	20 (47.6)	22 (52.4)	42 (100)	13.275 (0.001)
Never smoked	73 (23.5)	237 (76.5)	310 (100)	
Quit smoking	18 (37.5)	30 (62.5)	48 (100)	
Number of years working with pesticides on farm				
1–5	35 (21.0)	132 (79.0)	167 (100)	7.743 (0.052)
6–10	51 (32.9)	104 (67.1)	155 (100)	
11–15	21 (35.0)	39 (65.0)	60 (100)	
16–20	4 (22.2)	14 (77.8)	18 (100)	
Ever attended training on pesticide application				
Yes	16 (12.1)	116 (87.9)	132 (100)	24.002 (<0.001)
No	95 (35.4)	173 (64.6)	268 (100)	
Frequency of pesticide application per growing season				
Once	3 (23.1)	10 (76.9)	13 (100)	23.958 (<0.001)
Twice	27 (15.6)	146 (84.4)	173 (100)	
Thrice	47 (36.7)	81 (63.3)	128 (100)	
More frequently	34 (39.5)	52 (60.5)	86 (100)	
Knowledge category				
Good knowledge	53 (18.0)	241 (82.0)	294 (100)	52.310 (<0.001)
Poor knowledge	58 (54.7)	48 (45.3)	106 (100)	
Attitude category				
Negative attitude	78 (50.6)	76 (49.4)	154 (100)	65.492 (<0.001)
Positive attitude	33 (13.4)	213 (86.6)	246 (100)	

*p*-value < 0.05 indicates significance; Fisher’s exact test.

**Table 5 ijerph-22-00211-t005:** Ordinary logistic regression analysis of the respondents’ socio-demographic characteristics and category of pesticide handling and application practices.

Characteristics and Knowledge	ß	Sign.	Exp(ß)	Lower Bound	Upper Bound
Education					
No formal education	R.C	R.C	1.000	R.C	R.C
Completed primary school	0.856	0.032 *	2.355	1.097	5.054
Completed secondary school	0.534	0.016 *	2.445	1.184	5.047
Tertiary	2.090	<0.0001 *	8.082	2.626	24.870
Smoking Habit					
Quit smoking	R.C	R.C	1.000	R.C	R.C
Smokers	0.337	0.500	1.401	0.527	3.726
Never smoked	0.691	0.062	1.995	0.966	4.121
Ever attended training on pesticide application					
No	R.C	R.C	1.000	R.C	R.C
Yes	1.037	0.001 *	2.821	1.513	5.261
Frequency of pesticide application per growing season					
Once	R.C	R.C	1.000	R.C	R.C
Twice	0.182	0.812	0.833	0.186	3.730
Thrice	1.067	0.159	0.344	0.078	1.517
More frequently	0.962	0.210	0.382	0.085	1.721
Knowledge category					
Poor	R.C	R.C	1.000	R.C	R.C
Good	1.704	<0.001 *	5.494	3.385	8.919
Attitude Category					
Negative attitude	R.C	R.C	1.000	R.C	R.C
Positive attitude	1.891	<0.001 *	6.624	4.083	10.748

* Significant at 5%. Note: R.C = Reference Category.

## Data Availability

The raw data supporting the conclusions of this article will be made available by the authors upon request.

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
