# Peer review of "Determinants of Safe Pesticide Handling and Application Among Rural Farmers"

_ijerph, 2025, doi:10.3390/ijerph22020211_

Round 1
Reviewer 1 Report (Previous Reviewer 2)
Comments and Suggestions for Authors
Most questions have been modified. But still need more revision.
1. Line 150, “rregularly” is a wrong word.
2. The literature format needs to be standardized. The references should be consistent each other.
Author Response
Dear Reviewer
Many thanks for the comments.
We have responded to each points raised accordingly.
|
|
Line 150, “rregularly” is a wrong word. |
Many thanks for brining this error to our attention. This has now been updated accordingly |
The literature format needs to be standardized. The references should be consistent each other. |
Many thanks for the comment we have now gone through ensure consistency as suggested |
With Best regards
Reviewer 2 Report (Previous Reviewer 3)
Comments and Suggestions for Authors
Although the authors have been generally responsive to comments on the initial submission, the statement of limitations that the authors have added is inadequate and contains grammatical errors. They state, “The authors acknowledge the study was limited to two local government areas and 379 thereby calls for caution around the generalization of farmers safety behavior in Oyo State and Nigeria at Large. In addition, considering the vegetation.”
Other limitations include the cross-sectional nature of the research.
Grammatical errors: (1) “farmers safety behavior” should be “farmers’ safety behaviors”; (2) “In addition, considering the vegetation” is not a sentence.
Author Response
Dear Reviewer
Many thanks for the comments.
We have responded to each points raised accordingly.
Other limitations include the cross-sectional nature of the research. |
Many thanks for your comment. The limitation has been updated to reflect what was stated here. |
Grammatical errors: (1) “farmers safety behavior” should be “farmers’ safety behaviors”; (2) “In addition, considering the vegetation” is not a sentence. |
Authors acknowledge this typo errors. Related section now updated |
In addition, considering the vegetation.” |
The incomplete sentence has now been deleted |
With thanks
Haruna
This manuscript is a resubmission of an earlier submission. The following is a list of the peer review reports and author responses from that submission.
Round 1
Reviewer 1 Report
Comments and Suggestions for Authors
General Comments:
Extensive English language editing is required throughout the manuscript
Introduction:
Okay, sufficient details were provided, however it could be further improved by adding a discussion of methods used in similar studies and the rationale behind their selection.
Methods:
Justifications may be required for the following:
Excluding younger farmers (<18 years) and those with less experience might overlook insights from emerging farmers or those starting to engage with pesticides.
The population size of 100,000 active farmers is a "guesstimate" rather than based on verified records, introducing uncertainty into the sample size calculation.
This could lead to under-sampling or over-sampling, affecting the representativeness of the finding
Uncertainty, potential biases, and limitations of the methods and how the authors attempted to alleviate them may be described.
Results
Table 2. Digits shall be in front of the respective questions.
Table 5. Model fit statistics may be provided
The study does not consider confounding factors that may influence the predictors and the outcomes i.e. access to info, access to training, extension services etc
In the discussion section, the limitations of the study and how they could be alleviated must be discussed
Conclusions:
Include a summary of your main findings, identify gaps for future research
Comments on the Quality of English Language
English language editing is required throughout the manuscript i.e. spelling, grammar and syntactical errors may be removed
Author Response
Dear Reviewer
Many thanks for the comment/observation made to our submitted manuscript. Your comments has helped improve the quality of the manuscript and authors appreciate your time spent on the task. Find attached our response to each comments raised.
With thanks
Haruna

Reviewer 2 Report
Comments and Suggestions for Authors
Review comments on ijerph-3314091:
The study investigated the determinants of safe pesticide handling and application among farmers at two sites in Nigeria. The result is meaningful to certify the necessity of formulating relevant management policies and rules. The method taken was scientific and seriously. But the writing is rough and still needs polishing. The literature format needs to be standardized. Some more suggestions are as follows.
Line 53,“are classed according to WHO as highly hazardous group” are suggested to be revised to “are classed as highly hazardous group according to WHO”.
Table 1, last line 6-9. “1-5yers, 6-10years, 11-15years, 16-20years” is suggested to be changed to “1-5 years, 6-10 years, 11-15 years, 16-20 years”.
Line 180, change “disagreed on with washing” to “disagreed on washing”?
Line 182-183, change “sprayer thanks” to “sprayer tanks”?
Line 188, change “due its convenience” to “due to its convenience”?
Line 195, change “the they claim” to “they claim”?
Line 204, change “with mouth to to” to “with mouth to”.
Page 6-7, table 3, line 6-7, change “face and hand” to “face and hands”?
Line 211, “Figure 1: Category of Pesticide handling and application Practices” is suggested to be changed into “Figure 1: Health Influence of incorrectly pesticide handling and application”?
Line 227-232, “Survey of most commonly used pesticides among the respondents revealed 54.8% preferred Cypermethrin (trade name “Best”) a WHO class II pesticide was the most reported as preferred chemical of choice. Another WHO Class II pesticide Lamda-cyhalothrin (Lara Force) is used among 37.9% of the respondent while Dichlorvos (Sharpshooter) WHO Class I pesticide another 2.8% of the respondents confirmed its use on their farms.” The sentences are hard to be understood, how about “Survey of most commonly used pesticides among the respondents revealed that 54.8% of respondents preferred Cypermethrin (trade name “Best”), a WHO class II pesticide, which was reported as the most preferred chemical of choice. Another WHO Class II pesticide Lamda-cyhalothrin (Lara Force) is used among 37.9% of the respondent. While Dichlorvos (Sharpshooter), WHO Class I pesticide was confirmed to be used by other 2.8% of respondents on their farms.”?
Line 240-241, “Having participants who have had training on pesticide application and use presented better safe pesticide handling practices during and after usage”, unclear.
In Table 5, first line, what’s the meaning of “upper ” and “lower”?
Line 271, change “behind their inability of to read” to “behind their inability to read”?
Line 278, “and or”?
Line 283, change “in line with” to “In line with”.
Line 318, change “Southwestern Nigeria” to “southwestern Nigeria”?
Line 361-362, “For research articles with several authors, a short paragraph specifying their 361 individual contributions must be provided. The following statements should be used” deleted?
Comments on the Quality of English LanguageReview comments on ijerph-3314091:
The study investigated the determinants of safe pesticide handling and application among farmers at two sites in Nigeria. The result is meaningful to certify the necessity of formulating relevant management policies and rules. The method taken was scientific and seriously. But the writing is rough and still needs polishing. The literature format needs to be standardized. Some more suggestions are as follows.
Line 53,“are classed according to WHO as highly hazardous group” are suggested to be revised to “are classed as highly hazardous group according to WHO”.
Table 1, last line 6-9. “1-5yers, 6-10years, 11-15years, 16-20years” is suggested to be changed to “1-5 years, 6-10 years, 11-15 years, 16-20 years”.
Line 180, change “disagreed on with washing” to “disagreed on washing”?
Line 182-183, change “sprayer thanks” to “sprayer tanks”?
Line 188, change “due its convenience” to “due to its convenience”?
Line 195, change “the they claim” to “they claim”?
Line 204, change “with mouth to to” to “with mouth to”.
Page 6-7, table 3, line 6-7, change “face and hand” to “face and hands”?
Line 211, “Figure 1: Category of Pesticide handling and application Practices” is suggested to be changed into “Figure 1: Health Influence of incorrectly pesticide handling and application”?
Line 227-232, “Survey of most commonly used pesticides among the respondents revealed 54.8% preferred Cypermethrin (trade name “Best”) a WHO class II pesticide was the most reported as preferred chemical of choice. Another WHO Class II pesticide Lamda-cyhalothrin (Lara Force) is used among 37.9% of the respondent while Dichlorvos (Sharpshooter) WHO Class I pesticide another 2.8% of the respondents confirmed its use on their farms.” The sentences are hard to be understood, how about “Survey of most commonly used pesticides among the respondents revealed that 54.8% of respondents preferred Cypermethrin (trade name “Best”), a WHO class II pesticide, which was reported as the most preferred chemical of choice. Another WHO Class II pesticide Lamda-cyhalothrin (Lara Force) is used among 37.9% of the respondent. While Dichlorvos (Sharpshooter), WHO Class I pesticide was confirmed to be used by other 2.8% of respondents on their farms.”?
Line 240-241, “Having participants who have had training on pesticide application and use presented better safe pesticide handling practices during and after usage”, unclear.
In Table 5, first line, what’s the meaning of “upper ” and “lower”?
Line 271, change “behind their inability of to read” to “behind their inability to read”?
Line 278, “and or”?
Line 283, change “in line with” to “In line with”.
Line 318, change “Southwestern Nigeria” to “southwestern Nigeria”?
Line 361-362, “For research articles with several authors, a short paragraph specifying their 361 individual contributions must be provided. The following statements should be used” deleted?
Author Response

(The authors gave the same response as above.)

Reviewer 3 Report
Comments and Suggestions for Authors
The authors present an interesting and important analysis of the association of farmer personal characteristics and pesticide safety behaviors among Nigerian farmers from two local government areas. The design of the overall study appears to be appropriate, and the paper is generally presented in a clear and straightforward manner. The authors need to provide more detail and clarification on several points.
The authors should have the paper edited by a native English speaker to improve the clarity of their presentation.
The Introduction provides a justification for the research and analysis.
The discussion of the Methods requires further detail.
Section 2.1. Study Location: The descriptions of the Ido and Ibarapa LGAs is unbalanced, with much more detail about the Ibarapa LGA provided.
Section 2.2. Study Design and Study Population: While this section includes some of the inclusion criteria, it lack a detailed statement of the study design and statistical power. The use of the term “guesstimate” is inappropriate.
Section 2.3. Sampling Procedure: This section needs more detail to answer three questions. (1) What were the two staged used in stratifying the ample? (2) How were potential participants (farmers) identified? (3) What was the participation/refusal rate?
Section 2.4. Data Collection Tools Procedures: The authors need to clarify what they mean by a “semi-structured” questionnaire. Did the interviewers have the authority to change the question wording and order? They authors list some of the measures in the questionnaire, but not all; no measures are defined in the paper. No references for the different measures are provided.
Two points need to be clarified in the results.
(1) In Table 1 and on line and line 50, the authors note that only 10.5% of the participants are current smokers,, but on line 172 they note that 72% of the participants state that they smoke when applying pesticides.
(2) On line 215 the authors note that there is no association between participant age and duration of pesticide use. This does not conform to most other studies – how can an 18 year old have used pesticides for as long as a 46 year old?
The Discussion lacks a statement of study limitation.
Comments on the Quality of English LanguageThe English usage is understandable but awkward.
Author Response

(The authors gave the same response as above.)

Round 2
Reviewer 1 Report
Comments and Suggestions for Authors
The authors have made certain improvements to the manuscript. In other cases, their limitations are acknowledged. The results of the study are important for planning and policymaking and could serve as a baseline for future researchers.
Author Response
Dear Reviewer
Many thanks for your comments.
The authors have made certain improvements to the manuscript. In other cases, their limitations are acknowledged. The results of the study are important for planning and policymaking and could serve as a baseline for future researchers.
Response: We acknowledged your positive comment and will like to extend our appreciation for the time taken to review our revised manuscript
Yours sincerely
Haruna
Reviewer 3 Report
Comments and Suggestions for Authors
The authors still do not define/operationalize their measures in the presentation of methods.
Author Response
Dear Reviewer
Many thanks for the comment raised.
Response: It is not clear to us what exactly you want us to address. We have provided step by step approach to what was done to arrive at the result presented. We have cited previous study adopted etc.
The word operationalize/define used above seem vague as we have presented these steps.
Yours sincerely
Haruna